# Effects of Early-Life Stress, Postnatal Diet Modulation and Long-Term Western-Style Diet on Peripheral and Central Inflammatory Markers

**DOI:** 10.3390/nu13020288

**Published:** 2021-01-20

**Authors:** Silvie R. Ruigrok, Maralinde R. Abbink, Jorine Geertsema, Jesse E. Kuindersma, Nina Stöberl, Eline M. van der Beek, Paul J. Lucassen, Lidewij Schipper, Aniko Korosi

**Affiliations:** 1Brain Plasticity Group, Center for Neuroscience, Swammerdam Institute for Life Sciences, University of Amsterdam, 1098 XH Amsterdam, The Netherlands; s.r.ruigrok@uva.nl (S.R.R.); maralinde@live.nl (M.R.A.); j.i.geertsema@uva.nl (J.G.); jessekuindersma@gmail.com (J.E.K.); stoberln@cardiff.ac.uk (N.S.); p.j.lucassen@uva.nl (P.J.L.); 2Department of Pediatrics, University Medical Centre Groningen, University of Groningen, 9713 GZ Groningen, The Netherlands; e.m.van.der.beek@umcg.nl; 3Danone Nutricia Research, 3584 CT Utrecht, The Netherlands; Lidewij.Schipper@danone.com

**Keywords:** early-life stress, arcuate nucleus of the hypothalamus, western-style diet, adipose tissue, neuroinflammation, microglia

## Abstract

Early-life stress (ES) exposure increases the risk of developing obesity. Breastfeeding can markedly decrease this risk, and it is thought that the physical properties of the lipid droplets in human milk contribute to this benefit. A concept infant milk formula (IMF) has been developed that mimics these physical properties of human milk (Nuturis^®^, N-IMF). Previously, we have shown that N-IMF reduces, while ES increases, western-style diet (WSD)-induced fat accumulation in mice. Peripheral and central inflammation are considered to be important for obesity development. We therefore set out to test the effects of ES, Nuturis^®^ and WSD on adipose tissue inflammatory gene expression and microglia in the arcuate nucleus of the hypothalamus. ES was induced in mice by limiting the nesting and bedding material from postnatal day (P) 2 to P9. Mice were fed a standard IMF (S-IMF) or N-IMF from P16 to P42, followed by a standard diet (STD) or WSD until P230. ES modulated adipose tissue inflammatory gene expression early in life, while N-IMF had lasting effects into adulthood. Centrally, ES led to a higher microglia density and more amoeboid microglia at P9. In adulthood, WSD increased the number of amoeboid microglia, and while ES exposure increased microglia coverage, Nuturis^®^ reduced the numbers of amoeboid microglia upon the WSD challenge. These results highlight the impact of the early environment on central and peripheral inflammatory profiles, which may be key in the vulnerability to develop metabolic derangements later in life.

## 1. Introduction

Exposure to adversity in the perinatal period increases one’s risk of developing metabolic diseases, such as obesity, later in life. Metabolic diseases are increasingly prevalent in modern society. According to the World Health Organization, 50% of people in the European Union are overweight or obese. Moreover, the number of people adopting unhealthy “Western-style” diets has risen [1]. These diets are characterized among others by a higher intake of (saturated) fats, and known to contribute to obesity development [2]. Understanding the (neuro)biological substrates that increase the risk of developing metabolic diseases is thus important.

There is emerging evidence that the perinatal environment defines the set-point around which bodyweight is regulated later in life [3,4]. Both the brain and adipose tissue rapidly develop in this period, making them sensitive to environmental disturbances [5,6]. Exposure to stress (e.g., physical, sexual or emotional abuse) during the early postnatal period increases the risk of developing obesity and metabolic syndrome later in life [7,8,9]. Importantly, the metabolic alterations induced by early-life stress (ES) are modulated by later-life diet: diets with a higher fat content during adolescence and adulthood can exacerbate adiposity in rodents exposed to ES induced by either limiting the nesting and bedding material or maternal separation in the early postnatal period [10,11].

Perinatal nutrition also modulates offspring risk of becoming obese, as shown in clinical and preclinical studies [12,13,14,15]. Adequate nutrition during development not only provides the necessary energy, but also the required building blocks for the developing offspring. Human milk offers optimal nutrition during development, and breastfeeding has been shown to have many health benefits including a lower risk for childhood obesity, diabetes type 2 and infection [16,17,18,19,20]. Importantly, the effects of breastfeeding on overweight and obesity last into adulthood [19], and early nutrition thus appears to “program” aspects of later metabolism.

Human milk contains nutrients as well as other bioactive substances, including hormones and cytokines that may contribute to these health benefits [16]. Another important characteristic of human milk is the physical property of its lipid droplets [21]. The lipid droplet of human milk is large and covered by a phospholipid trilayer consisting of membrane proteins, phospholipids and cholesterol (milk fat globule membrane; MFGM). In contrast, the lipid droplets that make up regular infant formula (IMF) are much smaller and lack the complexity of the surface area characteristics [22]. To resemble these physical properties of human milk more closely, a novel concept IMF was developed (Nuturis^®^, N-IMF), which contains larger lipid droplets that are surrounded by phospholipids. Interestingly, an early-life nutritional intervention with Nuturis^®^ in mice attenuated Western-style diet (WSD)-induced fat accumulation in adulthood up to 12 weeks of WSD exposure [23,24,25]. These protective effects disappeared after more prolonged exposure to WSD (27 weeks) [26], as well as upon 18 weeks of exposure to a more severe high-fat diet (HFD, with a higher fat content compared to WSD) [27].

How the early life environment determines energy homeostasis later in life is unknown. Maintenance of energy homeostasis is regulated by a complex interplay between the body and the brain. The orexigenic and anorexigenic neuronal populations in the arcuate nucleus of the hypothalamus are the main brain circuits involved in the regulation of energy intake and expenditure and its structural development continues into early postnatal life in rodents [5]. ES and early-life dietary lipid quality have both been shown to alter the structural development of these hypothalamic feeding circuits [28,29], which may possibly contribute to the susceptibility to metabolic disease.

Next to these important hypothalamic neuronal populations and their neuropeptides, the role of peripheral and central inflammation has received increasing attention in the contexts of high-fat feeding and obesity. Obesity results in macrophage accumulation and inflammation in the adipose tissues, and causes a systemic low-grade inflammation, which may contribute to the health risks associated with obesity, like insulin resistance [30,31,32]. In addition, hypothalamic glia cells, including microglia, the brain’s resident immune cells, are involved in body weight homeostasis and obesity [33,34,35]. Importantly, hypothalamic inflammation and gliosis occur as soon as three days after high-fat diet feeding, and before changes in bodyweight occur, suggesting that hypothalamic inflammation has an important role in diet induced obesity (DIO) [36]. Indeed, blocking one of the inflammatory pathways in the hypothalamus attenuated weight gain upon HFD feeding [37], and specifically modifying microglia mitochondrial functions could protect against HFD-induced weight gain [38]. Hypothalamic inflammation may thus be an essential mechanism contributing to the development of DIO.

Interestingly, there also seems to be a role for (neuro)inflammation in ES-induced programming. For example, systemic inflammation occurs in individuals exposed to early stress [39,40,41,42], and perinatal stress lastingly alters microglia in the hippocampus, as well as their response to challenges later in life [43,44,45,46]. Whether ES also affects hypothalamic microglia, however, is still unknown.

The mechanisms underlying how ES and postnatal nutrition modulate obesity risk, are currently poorly investigated. Given the reported detrimental effect of ES and the protective effects of Nuturis^®^ on later life metabolic outcomes [10,23,24,25], we have previously investigated the effects of these environmental manipulations, both independently and in combination, on the metabolic phenotype [26]. We did so under standard dietary conditions later in life, as well as upon exposure to WSD. Instead of using the more commonly used HFD exposure for a few days or weeks, we specifically chose a prolonged exposure (27 weeks) to WSD, with a more moderate fat content, to better resemble human dietary exposure. We observed some interesting dynamics in body fat accumulation during (early) adulthood, which appeared to be transient: by the end of the prolonged exposure to WSD, ES and Nuturis^®^ did no longer modulate fat accumulation [26].

Considering the increasing interest in the role of peripheral and central inflammation in metabolic derangements and obesity, we here investigated short and long-term effects of ES and Nuturis^®^ on the adipose tissue and hypothalamic immune profile: (1) directly after ES, i.e., at postnatal day (P) 9, (2) in adolescence, i.e., at P42 (adipose tissue only), and (3) in adulthood, both (a) under standard adult diet conditions, as well as (b) in response to a prolonged WSD. We used the same cohort of adult mice as in our previous study [26], which allowed us to relate our current findings to the available detailed characterization of their metabolic phenotype.

## 2. Materials and Methods

### 2.1. Animals and Breeding

8-week-old female and 6-week-old male C57Bl6j mice were ordered from Envigo (Venray, The Netherlands). Upon arrival, the mice were housed in groups of 3–5, and allowed to acclimatize for 2 weeks prior to breeding under standard housing conditions (type 3 conventional cage, sawdust bedding, paper straw cage enrichment, temperature: 20–22 °C, humidity: 40–60%, standard 12/12 h light/dark schedule (lights on 08:00 a.m.), food and water ad libitum). Breeding was performed in house to standardize the perinatal environment. To allow for mating, one male was housed with two nulliparous females for one week. Females were housed in pairs for another week in a clean cage, after which pregnant females were single-housed in a cage with filtertop equipped with standard bedding and nesting material (type 2 conventional cage, sawdust, one square piece of cotton nesting material (5 × 5 cm; Technilab-BMI, Someren, The Netherlands)). Birth of pups was monitored every 24 h before 09:00 a.m. When litters were born before 09:00 a.m., the previous day was designated as postnatal day (P) 0. Animals were weaned at P21 and housed with same-sex littermates (2–4 per cage). Only male offspring were used for these studies. Figure 1 shows an overview of the experimental groups and design. For P9 adipose tissue experiments, experimental groups were as follows: control (CTL): *n* = 9 (of 3 litters); ES: *n* = 7 (of 2 litters). For P9 microglia experiments, experimental groups were as follows: CTL: *n* = 7 (of 4 litters); ES *n* = 6 (of 3 litters). Experimental groups for P42 experiments were: CTL S-IMF: *n* = 7 (of 2 litters); ES S-IMF: *n* = 8 (of 3 litters); CTL N-IMF: *n* = 8 (of 3 litters); ES N-IMF: *n* = 8 (of 3 litters). For studies at P230, experimental groups were as follows: CTL S-IMF standard diet (STD): *n* = 12 (of 4 litters); CTL N-IMF STD: *n* = 12 (of 4 litters); ES S-IMF STD: *n* = 13 (of 3 litters); ES N-IMF STD: *n* = 11 (of 5 litters); CTL S-IMF WSD: *n* = 13 (of 4 litters); CTL N-IMF WSD: *n* = 13 (of 4 litters); ES S-IMF WSD: *n* = 12 (of 4 litters); ES N-IMF WSD: *n* = 12 (of 3 litters). Mice were kept under standard housing conditions.

All experiments were approved by the Animal Experiment Committee of the University of Amsterdam and complied with national legislation and the principles of good laboratory animal care following the European Union (EU) directive 2010/63/EU for the protection of animals used for scientific purposes.

### 2.2. Early-Life Stress (ES) Paradigm

From P2 to P9, animals were exposed to either control (CTL) or ES conditions. The limited nesting and bedding material model was used to induce ES, as previously described [10,47,48]. In short, on the morning of P2, litters were culled to 6 pups per litter (including both sexes, but when possible optimized for more males), weighted, and randomly assigned to CTL or ES condition. Litters with less than 5 pups or only containing one sex were excluded. CTL litters received standard amounts of sawdust bedding and a square piece of cotton nesting material (5 × 5 cm; Technilab-BMI, Someren, The Netherlands). ES cages were equipped with a fine-gauge stainless steel mesh, which was placed 1 cm above the cage floor. The cage floor was covered with a minimal amount of sawdust and dams received halve of the nesting material (2.5 × 5 cm). Cages were covered with filtertops. The litters were left undisturbed until the morning of P9, when the animals were weighted again and transferred to a clean cage with standard amounts of bedding and nesting material.

### 2.3. Experimental Diets

For the short-term (P9) studies, to investigate acute effects of ES, the diet was standard rodent chow (SDS CRM (P), 801722, 9% energy from fat, 22% energy from protein, 69% energy from carbohydrates). For the adolescent (P42) and adult (P230) studies, diets were semisynthetic (Ssniff-Spezialdiäten GmbH, Soest, Germany) with a macro- and micronutrient composition according to AIN-93-purified diets for laboratory rodents [49]. Throughout breeding and until P16, dams were fed the AIN-93G diet (10% energy from fat). Between P16 and P42, litters either received standard infant milk formula diet (S-IMF) or Nuturis^®^ IMF diet (N-IMF) [24]. Compared to the S-IMF diet, the lipid droplets in the N-IMF diet have altered physical characteristics (i.e., large, phospholipid-coated lipid droplets), due to altered processing and the addition of bovine MFGM-derived phospholipids. IMF diets consisted of 28.3% fat (fully derived from IMF) complemented with carbohydrates and protein to match the AIN-93G diet. To preserve lipid structure, IMF diets were freshly provided in the form of a dough ball on the cage floor on a daily basis. Until weaning (P21), pups were also able to drink milk from their mother. At P42, mice were switched to either standard AIN-93M diet (standard diet (STD); 16.7% energy from fat, fully derived from soy) or WSD (39.8% energy from fat, derived from soy (15%) and lard (85%)) until the end of the study (P230). Absolute food intake of STD and WSD was not different across experimental groups, while WSD animals had a higher caloric intake compared to STD animals [26].

### 2.4. Tissue Collection

Adult (P230) animals were fasted for 4 h prior to sacrifice. Animals that were used to study the short-term effects of ES at P9 as well as the long-term effects at P230, were anesthetized via intraperitoneal injection of pentobarbital (120 mg/kg Euthasol^®^). Next, a piece of adipose tissue was rapidly dissected and immediately frozen on dry ice. More specifically, the inguinal depot at P9 and the gonadal depot at P42 and P230 were used for this study. In fact, at P9, only the subcutaneous inguinal adipose tissue is abundantly present, while visceral adipose tissue depots still need to develop. The gonadal adipose depot belongs to the group of visceral adipose tissues, which are considered more metabolically active and thought to be implicated in metabolic disease [50]. Afterwards, animals were transcardially perfused with 0.9% saline, followed by perfusion with 4% paraformaldehyde in phosphate buffer (PB 0.1 M, pH 7.4). Dissected brains were post-fixed overnight and stored in PB with 0.01% sodium azide at 4 °C until further processing. Prior to slicing, brains were cryoprotected in 30% sucrose in 0.1 M PB. Frozen brains were sliced in 40 μm thick sections and stored in antifreeze solution (30% Ethylene glycol, 20% Glycerol, 50% 0.05 M PBS) at −20 °C. For the adolescent (P42) studies, adipose tissue was quickly isolated after rapid decapitation (no anesthesia used) and immediately frozen on dry ice. Fresh frozen tissue was stored at −80 °C. Notably, the adult (P230) animals are the same animals as used in the study of Abbink et al., 2020, and underwent additional procedures that are not mentioned here (e.g., behavior tests and dual-energy X-ray absorptiometry scans) [26].

### 2.5. Real-Time PCR

RNA was extracted from either inguinal adipose tissue (P9) or gonadal adipose tissue (P42 and P230). In short, adipose tissue was homogenized in TRIzol (Invitrogen, Carlsbad, CA, USA) and samples were centrifuged to remove excessive fat. Chloroform was added, samples were centrifuged and the aqueous phase containing RNA was removed. Clean RNA was obtained with an RNA clean and concentrator kit and DNAse I treatment (ZYMO, Irvine, CA, USA), and stored at −80 °C. Next, cDNA was made with SuperScript II Reverse Transcriptase (Invitrogen), and stored at −20 °C until further use.

Relative expression of genes involved in multiple inflammatory pathways was assessed by RT-PCR performed on an Applied Biosystems 7500 Real-time PCR system (Thermo Fisher Scientific, Waltham, MA, USA). Hot FirePol EvaGreen Mastermix (Solis Biodyne, Tartu, Estonia) was used together with 150 nM of gene specific forward and reverse primers and a 0.135-ng/µL cDNA template. All primers (Eurogentec, Seraing, Belgium, Table 1) were tested for efficiency prior to experimental use and accepted when efficiency was between 90% and 110%. Cycling conditions were as follows: 15 min polymerase activation at 95 °C and 40 cycles of replication (15 s at 95 °C, 20 s at 65 °C, and 35 s at 72 °C). Qbase+ software (Biogazelle, Ghent, Belgium) was used for relative quantification of gene expression by the ∆∆Ct method. Expression was normalized for two reference genes, which were not affected by experimental conditions and tested for stability in Qbase+.

### 2.6. Immunohistochemistry

Hypothalamic microglia were visualized with immunohistochemistry for ionized calcium binding adaptor molecule 1 (Iba1), a marker for microglia/macrophages. Free floating brain slices containing hypothalamus were washed (3 × 5 min) in 0.05 M tris buffered saline (TBS, pH 7.6) before a 15-min incubation in 0.3% H_2_O_2_ in TBS to block endogenous peroxidase activity. Next, sections were washed in TBS (3 × 5 min) and blocked in 1% bovine albumin serum (BSA) + 0.3% triton in TBS (blocking mix) for 30 min. Primary antibody incubation occurred in blocking mix at room temperature for 1 h, followed by an overnight incubation at 4 °C (rabbit anti-Iba1, Wako, 019-19741, 1:5000). The next day, sections were washed in TBS + 0.3% triton (3 × 5 min), and incubated in goat anti-rabbit biotinylated secondary antibody (Vector Laboratories, Burlingame, CA, USA, 1:500) in blocking mix for two hours. After another washing step, sections were incubated in avidin–biotin complex in 0.05 M TBS for 90 min (Vectastain elite ABC-peroxidase kit, Brunschwig Chemie, Basel, Switzerland, 1:800). After washing sections in 0.05 M tris buffer (TB), Iba1 positive cells were visualized with 0.2 mg/mL diaminobenzidine (DAB) and 0.01% H_2_O_2_ in 0.05 M TB. Sections were thoroughly washed and mounted on pre-coated glass slides (Superfrost Plus slides, Menzel, Thermo Scientific, Waltham, MA, USA), dehydrated and cover slipped.

### 2.7. Quantification of Iba1 Coverage and Cell Density

Quantification of Iba1 staining was done by an experimenter blinded for condition and diet exposure. Only animals that had intact arcuate nucleus (ARH) after sectioning were used. This resulted in CTL: *n* = 7 and ES: *n* = 6 for P9 studies, and CTL S-IMF STD: *n* = 7; CTL N-IMF STD: *n* = 8; ES S-IMF STD: *n* = 8; ES N-IMF STD: *n* = 7; CTL S-IMF WSD: *n* = 8; CTL N-IMF WSD: *n* = 11; ES S-IMF WSD: *n* = 9; ES N-IMF WSD: *n* = 10 for the adult studies. Iba1 stained tissue was imaged using a 10x objective on a Nikon Eclipse Ni-E microscope using Nikon Elements software. Both sides of the ARH in 2–3 (P9) or 3–4 (P230) different sections per animal were imaged between bregma −1.055 and −1.955, based on the Allen Brain Atlas (2011). A region of interest (ROI) was defined at each side of the ARH, which, depending on the bregma, was placed with either one or two corners close to the third ventricle.

For coverage analysis of both P9 and adult ARH, a fixed threshold was chosen to select stained Iba1 positive cells including their ramifications without selecting background. Next, the percentage of stained area within the ROI was calculated. Within the same ROI, cells were counted manually and divided into three categories based on their morphology (type 1–3). Type 1, amoeboid microglia present a rounded cell body which can have pseudopodia but no long ramifications. Type 2 ramified cells are intermediate forms; they have fewer ramifications than type 3 cells and their ramifications can be thicker. Type 3 microglia present many thin ramifications [51]. The microglia were classified based on careful visual appreciation by a trained and blinded experimenter. In adulthood, microglia morphology is considered to be an indication of their activation: roughly, amoeboid microglia are considered activated while fully ramified microglia are considered surveying [51,52]. However, these morphology-based functional indications should be taken with caution as functional changes can be observed in the absence of detectable morphological alterations [51,52]. In contrast, during development, amoeboid microglia are indicative of a more immature state whereas ramified microglia are indicative of a more mature state [51,53].

### 2.8. Statistical Analysis

Data are expressed as mean ± standard error of the mean (SEM). Data were analyzed using SPSS 25.0 (IBM software, Chicago, IL, USA), Graphpad Prism 6 (Graphpad software, La Jolla, CA, USA), and considered significant when *p* < 0.05. For statistical analysis of RT-PCR results log transformed values were used. Firstly, the data were tested on outliers with SPSS, and significant outliers were removed from the dataset before proceeding with data analysis. Next, assumptions for parametric analysis were tested. Subsequently, the appropriate statistical test was performed. The general strategy was as follows: for comparisons between CTL and ES conditions at P9, independent *t*-tests (or a non-parametric alternative) was used. For analyzing the effects of both condition (CTL, ES) and postnatal diet (S-IMF, N-IMF) at P42, 2-way ANOVA was used. For the adult studies, we used the following approach: we first performed an independent *t*-test (or non-parametric alternative) on CTL S-IMF STD and CTL S-IMF WSD groups allowing us to assess the direct effects of the WSD on peripheral and central inflammation markers under control conditions and place our findings in the context of the current literature on DIO and inflammation [36,38,54,55,56]. Next, a 2-way ANOVA was performed on either STD or WSD groups (with condition and postnatal diet as predictor variables) to study modulating effects of ES and postnatal diet under standard (STD) and challenged (WSD) circumstances. Post-hoc comparisons were done with Tukey’s post-hoc test. Because multiple animals from one litter were included in this study, we always tested for possible contributing effects of litter by performing mixed model analysis with litter as random factor. We did not find effects of litter to any of the assessed parameters.

## 3. Results

### 3.1. ES Exposure Alters the Peripheral and Central Inflammatory Profile at P9

We first investigated direct effects of ES exposure on the peripheral and central inflammatory profile. Expression of inflammatory genes in the inguinal adipose tissue was assessed at P9. Expression of CCL2 and CCR2 mRNA, genes important for immune cell infiltration, were unaffected by ES (CCL2: t(14) = 0.313, *p* = 0.759; CCR2: t(14) = 1.153, *p* = 0.268) (Figure 2A,B). ES reduced the mRNA expression of F4/80, indicative of lower macrophage numbers (U = 9, Z = −2.199, *p* = 0.029) (Figure 2C), while NLRP3 expression, involved in the inflammasome, was not affected by ES (t(14) = 0.385, *p* = 0.706) (Figure 2D), nor was the expression of cytokine TNFα (t(7.7) = −1.753, *p* = 0.119) (Figure 2E).

Microglia coverage, cell density and subtype composition were determined in a defined ROI in the arcuate nucleus of the hypothalamus (ARH) of control and ES mice at P9 (Figure 3A,B). The area covered by Iba1-positive immunostaining was not affected by ES exposure (t(11) = −1.466, *p* = 0.171) (Figure 3C). Iba1 cell density however, was increased in the ARH of ES mice (t(11) = −2.969, *p* = 0.013) (Figure 3D). Moreover, while the relative composition of the different microglia subtypes was not affected by ES (%type1: t(11) = −1.518, *p* = 0.157; %type2: t(11) = −0.748, *p* = 0.47; %type3: t(11) = 1.553, *p* = 0.149) (Figure 3E), ES animals had a higher absolute density of type 1 microglia (t(11) = −2.263, *p* = 0.045), with the densities of type 2 and type 3 microglia being unaffected (type 2/mm^2^: U = 14.5, Z = −0.492, *p* = 0.639; type 3/mm^2^: t(11) = −0.376, *p* = 0.714) (Figure 3F–H).

### 3.2. Effects of Early-Life Stress and Postnatal Diet on Adolescent Adipose Tissue Inflammatory Markers

Next, we examined whether ES alters adipose tissue gene expression in adolescence (P42), and whether this is modulated by postnatal diet. Both condition and postnatal diet did not affect expression of CCL2 (F_condition_(1) = 0.001, *p* = −0.976; F_diet_(1) = 0.237, *p* = 0.631; F_condition*diet_(1) = 0.312, *p* = 0.581), CCR2 (F_condition_(1) = 0.123, *p* = 0.728; F_diet_(1) = 1.354, *p* = 0.255; F_condition*diet_(1) = 1.179, *p* = 0.288), and F4/80 (F_condition_(1) = 0.527, *p* = 0.474; F_diet_(1) = 0.25, *p* = 0.621; F_condition*diet_(1) = 0.011, *p* = 0.918) (Figure 4A–C). Postnatal N-IMF diet increased the expression of NLRP3, a component of the inflammasome, with no further modulation by condition (F_condition_(1) = 1.367, *p* = 0.255; F_diet_(1) = 4.759, *p* = 0.04; F_condition*diet_(1) = 0.022, *p* = 0.883) (Figure 4D), while ES exposure, independent of postnatal diet, increased the expression of the proinflammatory cytokine TNFα (F_condition_(1) = 5.399, *p* = 0.028; F_diet_(1) = 0.201, *p* = 0.658; F_condition*diet_(1) = 0.009, *p* = 0.924) (Figure 4E).

### 3.3. Prolonged WSD Increases Inflammatory Markers in Adipose Tissue and Amoeboid Microglia Numbers in the Hypothalamus

We next questioned whether ES and postnatal Nuturis^®^ feeding (1) have any effects on peripheral and central inflammation in adulthood, and (2) if this is modulated by a long-term western-style diet (WSD). Previous studies reporting inflammation in DIO often used high-fat diet (HFD; 45–60% energy from fat) for several days or weeks. Instead, in our study we chose a long-term WSD, which is more moderate in fat content (39.8% calories from fat) for ±27 weeks, to better mimic human diets. We therefore first studied if long-term exposure to such a diet results in a proinflammatory phenotype. Long-term WSD did not affect adipose tissue expression of CCL2 (t(21) = −1.714, *p* = 0.101), nor CCR2 (t(22) = −1.37, *p* = 0.185), while it increased the expression of F4/80 (t(12,546) = −3.105, *p* = 0.009) (Figure 5A–C). Moreover, NLRP3 expression was unaffected (t(20) = −0.602, *p* = 0.554) by WSD exposure, whereas TNFα expression (t(20) = −2.379, *p* = 0.027) was increased (Figure 5D,E).

In the ARH, the effect of WSD on microglia coverage, cell density and subtype composition was quantified in a defined ROI (Figure 6A,B). WSD did not affect the area covered by Iba1 positive immunostaining, neither did it affect Iba1 positive cell density (coverage: U= 23, Z = −0.579, *p* = 0.613; cell density t(13) = −0.379, *p* = 0.711) (Figure 6C,D). However, WSD did alter microglia subtype composition as shown by relatively higher numbers of type 1 and type 2 microglia, while lowering the percentage of type 3 microglia (%type1: t(11) = −2.384, *p* = 0.036; %type2: t(11) = −3.274, *p* = 0.007; %type3: t(11) = 4.804, *p* = 0.001) (Figure 6E). In line with these observations, type 1 cell density was also increased by WSD (type 1/mm^2^: t(12) = −2.261, *p* = 0.043; type 2/mm^2^: t(12) = −2.126, *p* = 0.055; type 3/mm^2^: t(12) = 0.996, *p* = 0.339) (Figure 6F–H).

### 3.4. Effects of ES and Postnatal Diet on Adult Adipose Tissue Inflammatory Markers and Hypothalamic Microglia under STD and WSD

Long-term WSD appeared to upregulate the expression of inflammatory markers in the adipose tissue and increase the density of amoeboid microglia in the hypothalamus. We next questioned if and how early-life stress exposure and/or postnatal Nuturis^®^ diet modulate the inflammatory profile on the long-term when exposed to either standard diet (STD) or WSD. On STD, CCL2, CCR2, and F4/80 were not affected by either ES or postnatal Nuturis^®^ diet (CCL2: F_condition_(1) = 0.769, *p* = 0.385; F_diet_(1) = 2.773, *p* = 0.103; F_condition*diet_(1) = 1.218, *p* = 0.276; CCR2: F_condition_(1) = 3.703, *p* = 0.061; F_diet_(1) = 2.278, *p* = 0.138; F_condition*diet_(1) = 0.279, *p* = 0.60; F4/80: F_condition_(1) = 0.532, *p* = 0.47; F_diet_(1) = 0.882, *p* = 0.353; F_condition*diet_(1) = 3.546, *p* = 0.067) (Figure 7A–C). NLRP3 expression levels (inflammasome component) however, were lower in Nuturis^®^ fed animals, without further modulation by condition (F_condition_(1) = 3.712, *p* = 0.062; F_diet_(1) = 5.364, *p* = 0.026; F_condition*diet_(1) = 0.619, *p* = 0.436) (Figure 7D). Expression levels of TNFα were not modulated by either ES or Nuturis^®^ when fed STD (F_condition_(1) = 0.48, *p* = 0.493; F_diet_(1) = 0.11, *p* = 0.742; F_condition*diet_(1) = 2.259, *p* = 0.142) (Figure 7E).

On WSD, neither condition nor postnatal diet affected inflammatory gene expression in the adipose tissue (CCL2: F_condition_(1) = 0.611, *p* = 0.439; F_diet_(1) = 0.634, *p* = 0.43; F_condition*diet_(1) = 1.727, *p* = 0.196; CCR2: F_condition_(1) = 0.008, *p* = 0.928; F_diet_(1) = 1.928, *p* = 0.172; F_condition*diet_(1) = 0.959, *p* = 0.333; F4/80: F_condition_(1) = 0.567, *p* = 0.456; F_diet_(1) = 0.995, *p* = 0.324; F_condition*diet_(1) = 0.093, *p* = 0.762; NLRP3: F_condition_(1) = 0.447, *p* = 0.507; F_diet_(1) = 0.1399, *p* = 0.244; F_condition*diet_(1) = 0.191, *p* = 0.665; TNFα: F_condition_(1) = 0.028, *p* = 0.869; F_diet_(1) = 0.314, *p* = 0.578; F_condition*diet_(1) = 1.243, *p* = 0.271) (Figure 7F–J).

Finally, we investigated the effects of ES and postnatal diet on microglia coverage, cell density and subtype composition under either STD or WSD conditions, in a defined ROI in the ARH (Figure 8A). On STD, none of the studied hypothalamic microglia parameters were significantly affected by either early-life condition or postnatal diet. The area covered by Iba1 positive immunostaining (F_condition_(1) = 0.091, *p* = 0.765; F_diet_(1) = 2.829, *p* = 0.105; F_condition*diet_(1) = 0.19, *p* = 0.666), nor Iba1-positive cell density (F_condition_(1) = 0.041, *p* = 0.841; F_diet_(1) = 0.662, *p* = 0.423; F_condition*diet_(1) = 0.318, *p* = 0.578) were affected by condition or postnatal diet (Figure 8B,C). In addition, relative subtype composition (%type1: F_condition_(1) = 0.156, *p* = 0.696; F_diet_(1) = 0.005, *p* = 0.943; F_condition*diet_(1) = 0.017, *p* = 0.898; %type2: F_condition_(1) = 0.003, *p* = 0.959; F_diet_(1) = 1.791, *p* = 0.193; F_condition*diet_(1) = 0.1941, *p* = 0.176; %type3 F_condition_(1) = 0.038, *p* = 0.847; F_diet_(1) = 1.553, *p* = 0.225; F_condition*diet_(1) = 1.737, *p* = 0.2) (Figure 8D), and absolute subtype cell densities (type 1/mm^2^: F_condition_(1) = 0.177, *p* = 0.677; F_diet_(1) = 0.029, *p* = 0.866; F_condition*diet_(1) = 0.133, *p* = 0.719; type 2/mm^2^: F_condition_(1) = 0.355, *p* = 0.557; F_diet_(1) = 0.091, *p* = 0.765; F_condition*diet_(1) = 0.175, *p* = 0.679; type 3/mm^2^: F_condition_(1) = 0.061, *p* = 0.808; F_diet_(1) = 2.066, *p* = 0.163; F_condition*diet_(1) = 1.154, *p* = 0.293) (Figure 8E–G) were unaffected.

As described above (Figure 6), WSD increased type 1 (amoeboid) microglia density in control mice. Animals on WSD that were exposed to ES had increased Iba1 coverage compared to control animals on WSD, which was not further modulated by Nuturis^®^ (F_condition_(1) = 7.824, *p* = 0.008; F_diet_(1) = 1.178, *p* = 0.285; F_condition*diet_(1) = 0.072, *p* = 0.79) (Figure 8H). Total microglia density was not affected by ES or postnatal Nuturis^®^ diet (F_condition_(1) = 0.757, *p* = 0.39; F_diet_(1) = 0.153, *p* = 0.698; F_condition*diet_(1) = 0.037, *p* = 0.849) (Figure 8I), however, postnatal Nuturis^®^ diet reduced the percentage of type 1 cells on WSD (F_condition_(1) = 0.208, *p* = 0.651; F_diet_(1) = 6.119, *p* = 0.019; F_condition*diet_(1) = 0.108, *p* = 0.745) (Figure 8J). Furthermore, postnatal diet modulated ES-induced effects on relative microglia subtype composition, as shown by interaction effects for both %type2 and %type3 cells (%type2: F_condition_(1) = 1.234, *p* = 0.275; F_diet_(1) = 0.926, *p* = 0.343; F_condition*diet_(1) = 5.121, *p* = 0.03; %type3 F_condition_(1) = 1.339, *p* = 0.256; F_diet_(1) = 0.008, *p* = 0.931; F_condition*diet_(1) = 4.531, *p* = 0.041) (Figure 8J). Tukey post-hoc analysis however revealed no significant differences between individual groups for both the percentage of type 2 and type 3 cells. In accordance with the reduced percentage of type 1 cells in Nuturis^®^ exposed animals, type 1 microglia density was affected by postnatal diet independent of condition (F_condition_(1) = 0.017, *p* = 0.897; F_diet_(1) = 4.698, *p* = 0.037; F_condition*diet_(1) = 0.13, *p* = 0.72) (Figure 8K). Type 2 and type 3 microglia density were not affected by condition or postnatal diet (type 2/mm^2^: F_condition_(1) = 0.007, *p* = 0.935; F_diet_(1) = 0.619, *p* = 0.437; F_condition*diet_(1) = 1.378, *p* = 0.249; type3/mm^2^ F_condition_(1) = 1.451, *p* = 0.237; F_diet_(1) = 0.528, *p* = 0.472; F_condition*diet_(1) = 2.291, *p* = 0.139) (Figure 8L,M).

## 4. Discussion

In the current study, we investigated effects of ES and postnatal diet on peripheral and central inflammatory markers, both in the short-term and into adulthood. We tested these parameters under standard dietary conditions as well as upon exposure to prolonged WSD in adulthood, the latter causing a mild peripheral and central inflammatory phenotype. We show that early in life, both directly after stress exposure as well as in adolescence, ES altered peripheral inflammatory gene expression, and that postnatal Nuturis^®^ diet modulated adipose tissue inflammatory gene expression in an age-dependent manner, independent of ES exposure. In addition, both ES and postnatal Nuturis^®^ diet had a lasting effect on the central inflammatory profile. We further observed increased microglia cell density and more amoeboid microglia in the hypothalamus directly after ES exposure. Interestingly, both ES and Nuturis^®^ modulated the inflammatory profile in adulthood upon prolonged WSD exposure. More specifically, WSD induced an increase in the number of amoeboid microglia, whereas postnatal diet with Nuturis^®^ reduced these numbers following WSD exposure. Furthermore, ES-exposed animals exhibited a different microglia subtype composition in response to WSD.

We will first discuss our findings on the effects of prolonged WSD on peripheral and central inflammation, and thereafter the modulatory effects of both ES and Nuturis^®^, upon STD and WSD exposure.

### 4.1. Prolonged WSD Exposure Leads to a Mild Inflammatory Phenotype in Control Animals

We earlier reported that prolonged WSD exposure increases fat accumulation [26]. In the current study, we show in the same mice that prolonged WSD also affects the peripheral and central inflammatory phenotype. We are the first to describe the inflammatory effects of such a prolonged WSD. An important difference between our study and the available literature is the amount of fat in the diet and the duration of exposure. In DIO studies, HFD is commonly used which contains 45–60% calories from fat, while exposure is typically shorter, i.e., several days to weeks. In our design, however, we used a prolonged (27 week) exposure to WSD, consisting of a more moderate fat content (39.8% energy from fat), which we believe better resembles dietary patterns of obese individuals.

Due to the limited literature with a similar design, we compared our findings to existing literature on the effects of HFD exposure [36,38,54,55,56,57]. Concerning peripheral inflammation, we observed an increase in the expression of the macrophage marker F4/80 and the cytokine TNFα in the adipose tissue, with no effects on the expression of the other examined inflammatory genes (CCL2, CCR2 and NLRP3) after WSD. Indeed, obesity and the expansion of adipose tissue have been described to increase M1 “classically activated” macrophage infiltration as well as the production of pro-inflammatory cytokines [30,58,59], which is in line with the increased F4/80 and TNFα expression upon WSD in our current study. In addition, in contrast to our observation, HFD has been described to increase the expression of the cytokine CCL2 [55,56]. CCL2 guides monocytes from the circulation to become tissue macrophages, which is the first step in the initiation of inflammation [60,61]. Although speculative, it is possible that due to the more prolonged nature of our (moderate) WSD exposure, after 27 weeks, new inflammatory cells are no longer recruited to the adipose tissue.

Concerning the central inflammatory profile, we observed a WSD-induced increase in amoeboid microglia numbers in the hypothalamus, with no effects on total microglia numbers or Iba1 coverage. Amoeboid types of microglia are considered to be activated, they secrete pro- and anti-inflammatory cytokines, and are thought to have a high phagocytic capacity for clearing debris and apoptotic cells [62]. Thus, the WSD-induced increase in the numbers of these amoeboid/activated microglia in the ARH would suggest an inflammatory phenotype. However, these conclusions are for now only based on morphological analyses of microglia. To gain further understanding of the potential functional alterations in microglia and the inflammatory state within the brain in response to WSD, it will be important to also test cytokine levels in the hypothalamus in the future. While this is in line with the direction of effects in response to HFD, an exposure to HFD for several days up to 16 weeks, a shorter period compared to our prolonged WSD, appears to have more marked effects on hypothalamic inflammation, including increases in microglia density and size [36,38,57].

Thus, even though the WSD clearly had some inflammatory effects, the differences in study design (fat percentage in diet, length of exposure, specifics of the control diet [63]) may explain some discrepancies with earlier literature. This highlights the importance of comparing different dietary fat percentages and durations of exposure with respect to the inflammatory processes they trigger, and how these may contribute to obesity.

### 4.2. Effects of ES and Nuturis^®^ on Peripheral Inflammation under STD and Prolonged WSD Exposure

ES leads to a reduction in F4/80 expression in the adipose tissue at P9, indicative of lower macrophage numbers. As shown in our previous study with these same animals, ES at this age also leads to a decrease in bodyweight [26], which was accompanied before by a reduction in fat mass [10]. It is intriguing to speculate about the biological mediators responsible for these changes in the adipose tissue inflammatory gene expression profile. Glucocorticoids (GCs) are a plausible candidate. In fact, ES leads to an increase in circulating GCs at P9 [47,48] and adipose tissue abundantly expresses the glucocorticoid receptor, which is not affected by ES [10]. GCs have been described to alter lipid metabolism, regulate adipogenesis, and increase central fat accumulation [64,65,66,67]. In addition, GCs are widely known for their anti-inflammatory capacity [68]. GCs inhibit adipocyte-induced macrophage recruitment in vitro, and dexamethasone (a synthetic GC) can prevent high-fat diet induced macrophage infiltration [69], which is further in line with our findings.

In adolescence (P42), ES induced a transient increase in TNFα expression in adipose tissue, which normalized in adulthood (P230). Exposure to ES has been shown to also increase circulating TNFα levels in adolescence [70], and we now show similar effects at the level of the adipose tissue. Moreover, regarding adipose tissue gene expression, prolonged WSD did not affect CTL and ES animals differently. As reported earlier, we neither observed effects of ES on adiposity, at either P42 or P230 after STD and WSD exposures [26]. One option is that the prolonged (±27 week) WSD challenge could possibly have overruled any subtle effects of ES in a similar way as it overruled ES effects on adiposity, which we observed after 8 weeks of WSD [10]. However, to the best of our knowledge, we are the first to study the effects of ES on adipose tissue inflammatory gene expression after a high-caloric diet, and there are no studies looking at shorter dietary interventions. As ES increases the risk of developing obesity, it will be key to understand in more detail how the adipose tissue responds to dietary challenges that contribute to adverse health risks.

Nuturis^®^ caused an initial increase in the expression of the inflammasome sensor NLRP3 in the adipose tissue in adolescence (P42), while decreasing its expression in adulthood (P230) under STD conditions, suggesting an age dependent and long-lasting effect of the postnatal diet. No such lasting effects were found after prolonged WSD. Inflammasomes are cytosolic multiprotein complexes that are part of the innate immune system, and stimulate the cleavage and secretion of cytokines. NLRP3 is an inflammasome sensor molecule that triggers the assembly of inflammasomes and is critical for their activation [71]. Alterations in NLRP3 expression levels may directly affect inflammasome levels, potentially changing the levels of cytokine released [72]. Importantly, NLRP3 expression in the adipose tissue has previously been associated with obesity-induced inflammation and insulin resistance in obese individuals, and diminishing NLRP3 levels can protect against insulin resistance in DIO mice [73]. This suggests that the Nuturis^®^-induced lower expression of NLRP3 in adulthood may provide some protective effects on insulin sensitivity on STD, which however now seems to be overruled by prolonged WSD. This is in line with earlier findings from our group showing that Nuturis^®^ lowered insulin resistance when fed an STD, but not a WSD [24,25]. NLRP3 expression may thus provide a mechanistic link between Nuturis^®^ and the earlier observed improved insulin sensitivity. How Nuturis^®^ might program NLRP3 expression needs to be further investigated. A possible hypothesis is that lipid metabolism may play a role; fatty acids have been shown to provide a priming signal for NLRP3 in vitro and after HFD feeding in mice [73,74,75], and Nuturis^®^ lastingly alters the plasma levels of certain lipids, including cholesterol and triglycerides [25].

### 4.3. ES and Nuturis^®^ Lastingly Affect Hypothalamic Microglia under STD and Prolonged WSD Exposure

We observed increased microglia cell density and an increase in amoeboid microglia in the arcuate nucleus of the hypothalamus directly after ES exposure at P9. Hypothalamic development continues into at least the first two weeks of postnatal life in rodents [5] and the microglia population expands and develops rapidly during this period [76,77,78,79]. Microglia change their morphology during development, which is commonly used to examine their maturity [53]. The amoeboid morphology suggests a more immature state and is in line with findings by us and others on the effects of ES in the hippocampus [43,45]. Moreover, the increased microglia density at P9 might be explained by a GC-triggered increase in microglia proliferation. Indeed, it has been shown that elevated GC levels, induced by several days of restraint stress, can stimulate microglia proliferation directly after the stress exposure [80]. However, opposite effects on microglia proliferation after only 24 h of in vitro GC exposure have also been described [81]. It is thus possible that the length of exposure to elevated GCs determines the observed effects on microglia proliferation.

In adult animals, we did not observe any ES effects on microglia under basal (STD) circumstances. However, when exposed to WSD, ES-exposed animals showed increased microglia coverage in the ARH of the hypothalamus. Because of the absence of significant differences in microglia subtypes between experimental groups, understanding the source of this increased coverage as well as its functional implication needs further investigation. However, in line with our finding, such programming effects of perinatal stress on microglia in response to later-life inflammatory challenges (e.g., Alzheimer pathology and bacterial infection) have been observed before both in the hippocampus and prefrontal cortex [43,44,82]. The literature on the effects of early adversity on hypothalamic microglia is sparse, and to the best of our knowledge, we are the first to show that ES also primes the hypothalamic microglia response to a WSD. Programming of hypothalamic microglia by early adversity has been described in a study concerning overfeeding: neonatally overfed rats showed increased microglia cell numbers and coverage when challenged with a bacterial infection [83]. We here describe a similar altered hypothalamic microglia response to a WSD challenge after ES exposure.

Interestingly, while prolonged WSD increased the number of amoeboid microglia, postnatal diet with Nuturis^®^ consistently reduced these numbers when fed WSD, independent of previous ES exposure. Moreover, Nuturis^®^ seemed to, in interaction with ES, modulate relative microglia subtype composition, although post-hoc analysis revealed no significant differences between individual groups. Nonetheless, these findings suggest Nuturis^®^ programs microglia and their response to later-life challenges. It is remarkable that a relatively subtle dietary intervention was able to modulate the microglial profile more than 6 months after the diet intervention was ended. Nuturis^®^ may impact microglia in multiple ways. The altered physical structure of lipids in Nuturis^®^ affects digestion and absorption kinetics and, as a result, the bioavailability of lipids essential for brain development [84]. Moreover, lipids are widely known for their capacity to modulate inflammation [85,86,87]. Both the type and quantity of fatty acids is known to affect microglia physiology directly as well as indirectly via hormones and the gut [88]. For example, gangliosides, which are present in Nuturis^®^, have been shown to have anti-inflammatory effects in microglia activated with lipopolysaccharide [89].

It is intriguing to speculate that Nuturis^®^ could modulate adipose tissue accumulation at least partly via microglia modulation. It is not clear yet how microglia could regulate fat accumulation, but one study showed that blocking microglia activation prevents HFD-induced fat accumulation and triggers the hypothalamic neurons that inhibit food intake, linking neuroinflammation to the neuronal populations that regulate energy homeostasis [38]. Indeed, blocking microglia activation led to increased energy expenditure and decreased food intake, preventing HFD-induced weight gain. Nuturis^®^ has been shown to modulate WSD-induced adiposity with WSD exposure up to 12 weeks [24,25], which was no longer present after a prolonged WSD exposure (27 weeks) [26], nor after 18 weeks of HFD exposure [27]. Further studies need to elucidate the complex involvement of hypothalamic microglia in the modulation of adipose tissue accumulation, and their modulation by ES and Nuturis^®^.

A limitation of our study is the lack of inclusion of female mice. As mentioned earlier, this study is the follow up of a large investigation, which encompassed both cognitive and metabolic readouts [26]. This original study was designed to include males as we had shown previously that the used ES model affects cognitive function and neurogenesis primarily in males [47], and also the reported effects of N-IMF on WSD-induced obesity have been studied in male mice [24,25], unfortunately limiting our possibility to include females in this study. However, there is increasing clinical and preclinical evidence that there are sex differences in the response to ES [11,90,91,92,93], early-life nutritional challenges [94], adipose tissue distribution and function [95], as well as in microglia [96,97,98], indicating the importance of studying the differential effects of ES and dietary manipulations in both males and females in future experiments.

## 5. Conclusions

Both ES and postnatal diet with Nuturis^®^ modulate peripheral and central inflammation. Even after prolonged WSD exposure that led to an obesogenic phenotype in all animals, both ES and Nuturis^®^ modulated markers of central inflammation. While more research is needed for a complete understanding, it is tempting to hypothesize that such effects might be involved in the earlier reported effects of both ES and Nuturis^®^ on adiposity and metabolic derangements [10,24,25]. These findings highlight the importance of the early-life environment on later life (metabolic) health.

## 6. Patents

The Nuturis intervention tested is a concept that is protected by several patents, all filed before the start of the experiments reported in this study.

## Figures and Tables

**Figure 1 nutrients-13-00288-f001:**
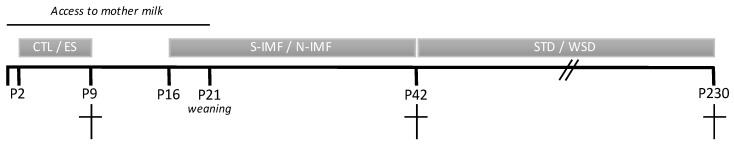
Experimental design. Animals were exposed to control (CTL) or early-life stress (ES) conditions from postnatal day (P) 2-P9. Postnatal diet (standard infant milk formula (S-IMF) or Nuturis^®^ IMF (N-IMF)) was provided from P16 to P42, and standard diet (STD) or western-style diet (WSD) from P42 to P230. Animals were sacrificed at either P9, P42, or P230.

**Figure 2 nutrients-13-00288-f002:**
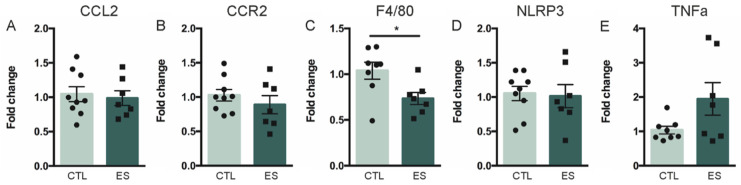
(**A**–**E**) effects of ES on adipose tissue inflammatory gene expression at P9. ES decreased expression of F4/80, a marker of macrophages. Data are presented as mean ± standard error of the mean (SEM); * effect of condition; *p* < 0.05.

**Figure 3 nutrients-13-00288-f003:**
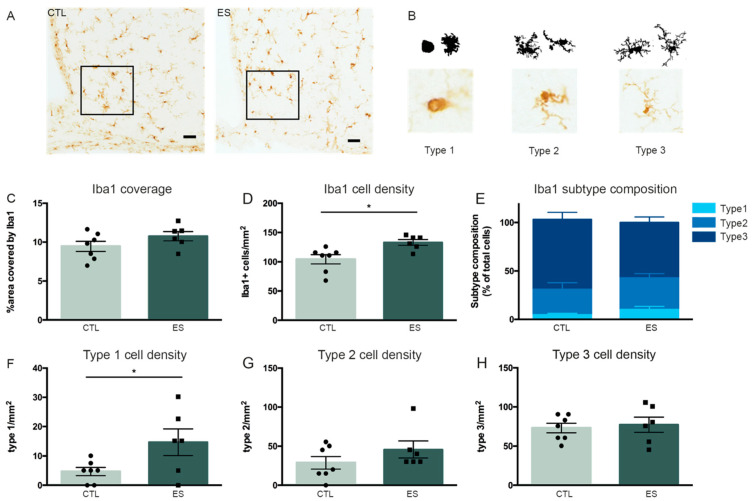
Microglia density is increased by ES at P9. (**A**) Representative images of Iba1 immunostaining and a defined region of interest (ROI; black square) in the arcuate nucleus of the hypothalamus (ARH). (**B**) Example illustrations of microglia subtypes. (**C**) Iba1 coverage is not affected by ES. (**D**) Iba1 cell density is increased directly after ES exposure. (**E**) ES does not alter Iba1 subtype composition at P9. (**F**–**H**) Iba1 subtype cell densities. ES increased type 1 cell density. Data are presented as mean ± SEM; scale bars 100 μm; * condition effect; *p* < 0.05.

**Figure 4 nutrients-13-00288-f004:**
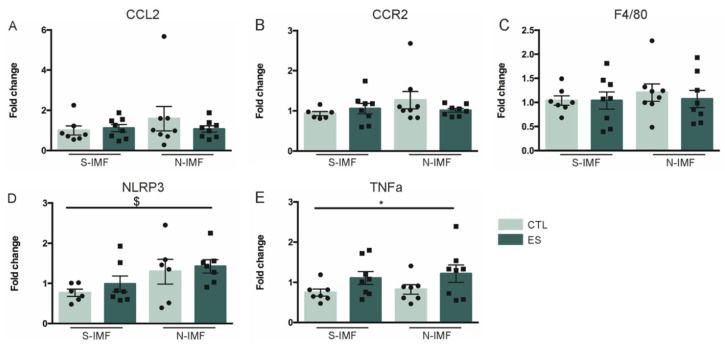
Effects of ES and postnatal diet on adipose tissue inflammatory gene expression at P42. (**A**–**C**) ES and postnatal diet do not affect CCL2, CCR2 and F4/80 expression. (**D**) N-IMF increased NLRP3 expression at P42. (**E**) ES increases TNFα expression levels. Data are presented as mean ± SEM; * condition effect; $ diet effect; *p* < 0.05.

**Figure 5 nutrients-13-00288-f005:**
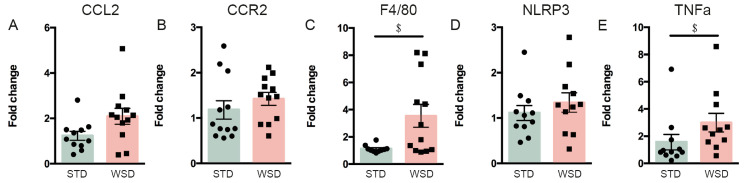
Effects of prolonged WSD exposure on adipose tissue inflammatory gene expression in CTL S-IMF animals at P230. (**A**,**B**,**D**) WSD exposure did not affect CCL2, CCR2 and NLRP3 expression levels. (**C**,**E**) WSD exposure increased F4/80 and TNFα expression in the adipose tissue. Data are presented as mean ± SEM; $ diet effect; *p* < 0.05.

**Figure 6 nutrients-13-00288-f006:**
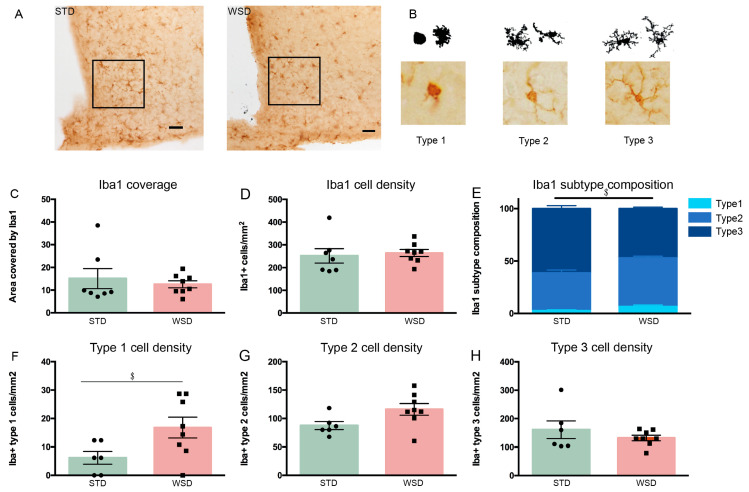
Effects of prolonged WSD exposure on hypothalamic microglia density and coverage. (**A**) Representative images of Iba1 immunostaining. Iba1 immunostaining was analyzed within a defined ROI (black square) in the ARH of STD- and WSD-exposed CTL S-IMF animals. (**B**) Example illustrations of microglia subtypes. (**C**) Iba1 coverage was not affected by WSD. (**D**) Iba1 cell density was not affected by WSD. (**E**) WSD altered Iba1 subtype composition at P230 by increasing the percentage of type 1 microglia. (**F**–**H**) Iba1 subtype cell densities. WSD increased type 1 cell density. Data are presented as mean ± SEM; scale bars 100 μm; $ diet effect; *p* < 0.05.

**Figure 7 nutrients-13-00288-f007:**
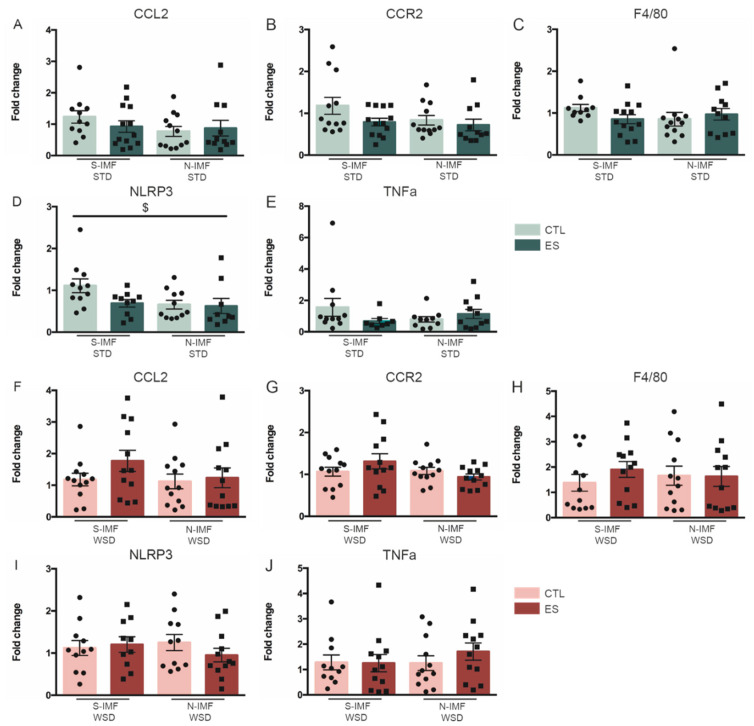
Effects of ES and postnatal diet on expression of inflammatory genes in adulthood, after either STD (**A**–**E**) or WSD (**F**–**J**). (**A**,**B**,**C**,**E**) ES nor postnatal diet affected adipose tissue gene expression of CCL2, CCR2, F4/80 or TNFα in adulthood when fed STD. (**D**) Postnatal N-IMF diet decreased NLRP3 expression at P230, when fed STD in adulthood. (**F**–**J**) ES and postnatal diet did not affect expression of CCL2, CCR2, F4/80, NLRP3 and TNFα in the adipose tissue when fed WSD in adulthood. Data are presented as mean ± SEM; $ diet effect; *p* < 0.05.

**Figure 8 nutrients-13-00288-f008:**
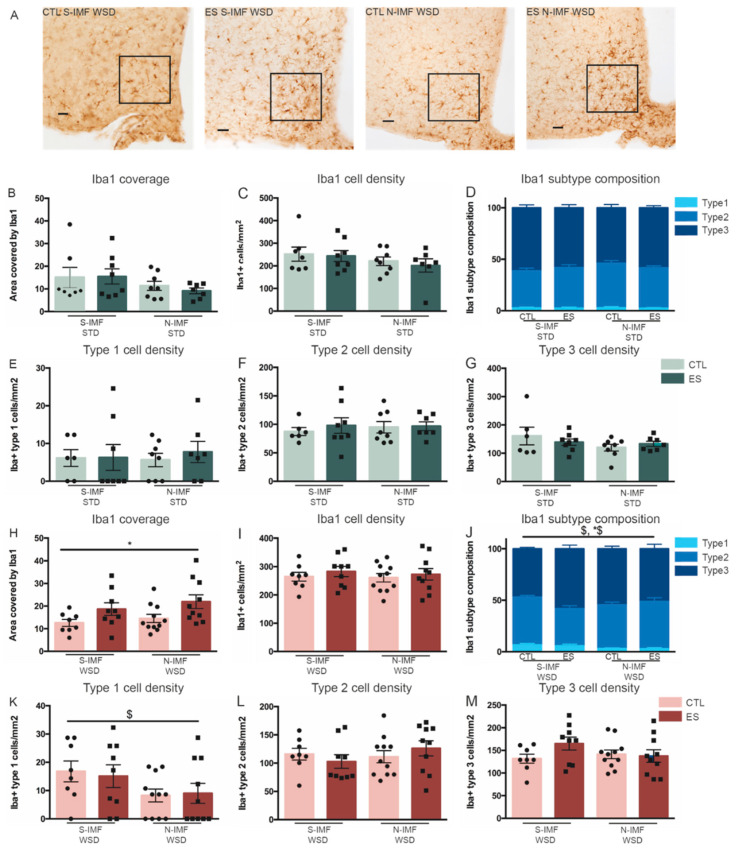
Effects of ES and postnatal diet on hypothalamic microglia at P230, when fed STD (**B**–**G**) or WSD (**H**–**M**) in adulthood. (**A**) representative images of Iba1 positive cells in the ARH of animals exposed to WSD. (**B**–**G**) On standard adult diet, ES nor postnatal diet affected hypothalamic microglia coverage, density or subtype composition (for different subtypes see Figure 3B and Figure 6B). (**H**) On WSD, ES exposure increased Iba1 coverage (**I**) On WSD, ES and postnatal diet did not affect Iba1 cell density. (**J**) Microglia subtype composition was affected by postnatal diet and ES in animals on WSD (%type 1 diet effect; %type 2 and %type 3 diet*condition effect). (**K**–**M**) Type 1 density was decreased by postnatal N-IMF diet. Data are presented as mean ± SEM; scale bars 100 μm; * condition effect; $ postnatal diet effect; *$ condition by postnatal diet effect; *p* < 0.05.

**Table 1 nutrients-13-00288-t001:** Primers used for RT-PCR.

Gene	Pathway/Function	Forward Primer (5′-3′)	Reverse Primer (5′-3′)
CCL2	Immune cell infiltration	AGCTGTAGTTTTTGTCACCAAGC	GTGCTGAAGACCTTAGGGCA
CCR2	Immune cell infiltration	AGGGAGACAGCAGATCGAGTG	ACAACCCAACCGAGACCTCTT
F4/80	Macrophage marker	TGTGTCGTGCTGTTCAGAACC	AGGATTCCCGCAATGATGG
NLRP3	Inflammasome	CAGCCAGAGTGGAATGACACG	GCGCGTTCCTGTCCTTGATA
TNFα	Cytokine	GTAGCCCACGTCGTAGCAAAC	AGTTGGTTGTCTTTGAGATCCATG
RPS29	Reference gene	AGTCACCCACGGAAGTTCGG	GTCCAACTTAATGAAGCCTATGTCCTT
RPL19	Reference gene	TTGCCTCTAGTGTCCTCCGC	CTTCCTGATCTGCTGACGGG

## Data Availability

Not applicable.

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
