# Peer review of "Effects of Early-Life Stress, Postnatal Diet Modulation and Long-Term Western-Style Diet on Peripheral and Central Inflammatory Markers"

_nutrients, 2021, doi:10.3390/nu13020288_

Round 1
Reviewer 1 Report
Ruigrok and colleagues that used an in vivo model of early life stress and postnatal diet manipulation in male mice and the effect on peripheral and central inflammation. They have shown that a somewhat subtle dietary intervention with Nuturis® altered inflammasome NLRP3 gene expression in adipose tissue in an age dependent manner with long-lasting effects. Moreover, Nuturis® modulates microglia and their responses to later-life challenges. The reviewer found the outcomes of this study interesting. However, there are still some reservations regarding this manuscript, which have been detailed below for the authors consideration.
Methods:
- 1 Animals and Breeding: More details are required on the mating to get a better understanding. Were both the males and females nulliparous rats?
- 2 Early-life stress paradigm: The authors state that the offspring were reduced to 6 pups per litter and only male pups were used for the duration of the experiment. Why were females not assessed in this study as there is a lot of literature to suggest that males and females have different responses to environmental stress and diets during development.
- 2 Early-life stress paradigm: Littermates share antenatal and postnatal environments thus are to some degree interdependent which can produce pervasive and persistent litter effects. Have the authors appropriately allocated offspring to experimental groups to include equal mice per group from different litters?
- 3 Experimental diets: Did the offspring receiving Nuturis® IMF diet consumed equivalent daily intake compared to standard milk formula diet?
- 4 Tissue Collection: The authors have stated that the adult (p230) animals were also used in the study of Abbink et al. 2019, however, the reference for this paper is Abbink et al. 2020. Please adjust this to be consistent with the reference.
- 7 Quantification of Iba-1 coverage and cell density: P5, line 223 states “For adult analysis …”, however, the authors do not provide any information regarding the Iba-1 analysis of the P9 tissue. I would assume this is the same for the P9 tissue, however, this should be included if it is different. The author’s also state that the microglial cells were divided into three categories i) amoeboid, ii) small ramifications or iii) fully ramified, however, the reviewer believes that more information regarding classification of the microglia is essential. Are the authors classifying small ramifications to have a certain number of primary processes compared to fully ramified which have more, if so, this should be stated.
Results:
- 1 ES exposure alterations the peripheral…: From this reviewer’s point of view, it would be beneficial to change the gene expression graph order to match the results section. For example, pg 6 line 260 discusses F4/80 first rather than CCL2 and CCR2 which are Fig 1a and Fig 1b.
- The microglia sections are quite extensive, often with 6+ graphs per figure, thus it would be helpful to properly state which graph in the figures the authors are referring to in the results sections, ie Fig 1A, B, etc.
- While most DIO investigations focus on the arcuate nucleus, as it is anatomically placed to receive nutrient signals from the peripheral circulation, feeding-related disturbances in the ARC are readily conveyed to other hypothalamic regions such as paraventricular, dorsomedial, lateral and ventromedical hypothalamic nuclei. It would be important, and strengthen the manuscript, to investigate the other hypothalamic regions in this model of early life stress, Nuturis and the long-term western-style diet.
- Please be mindful of the statistical output as there are often inconsistencies. CCL2 statistics appear to have an error on line 291 as the p value is -976. Line 308 discusses F4/80 gene expression between STD and WSD mice which is a t test and the degrees of freedom appears to be incorrect.
Discussion:
The discussion section is elegantly written and provides a detailed discussion of the project considering the current literature.
- It would be interesting to have a more detailed discussion as to the potential mechanisms behind the CCL2 mismatch between WSD and published HFD studies. Moreover, have the authors assessed T cell numbers in this early-life stress model?
- I would like to see a discussion on the limitations of the study due to the lack of female analysis as there is a plethora of literature of suggest that males and females have different responses to developmental disruptions.
Reviewer 2 Report
This manuscript represents an interesting and timely exploration of the impact of Nuturis, a concept infant milk formula, on early-life stress-induced changes in adipose tissue inflammatory gene expression and microglial number and morphology in the arcuate nucleus of the hypothalamus, as well as modulation of the response to a western-style diet. Although the study design is quite complex, the manuscript is overall well-written and the results clearly presented. I do have some issues with the interpretation of microglial morphology results and the statistical analyses employed that should be addressed prior to publication. All of my specific comments are listed below in the order that they appear in the manuscript:
- Methods, page 3, line 127: Please justify why only male offspring were used for these studies, as consideration of sex as a biological variable is an important component of study design.
- Methods, page 4, Figure 1: There are boxes above your timeline that are currently empty (may be a technical error), and that would be more helpful if they were labeled CTL/ES, S-IMF/N-IMF, STD/WSD.
- Methods, page 6, line 234: “type 1: amoeboid; type 2: small ramifications; type 3: fully ramified”. I have not heard of the “small ramifications” category before; do you mean stout microglia with few processes, or microglia with thick processes? Please cite the literature that you have referenced when determining these categories.
- Methods, page 6, lines 235-6: “Amoeboid microglia are considered activated, while ramified microglia are considered surveying.” These designations are only appropriate for adult brains, not for developing brains, where microglia progress through several developmental stages normally, from amoeboid morphology through ramified morphology, in the absence of any immune challenges. Thus, it is inappropriate to call these normally developing microglia “activated,” and they should instead be referred to by their actual morphology or in terms of developmental stages (immature vs. mature). In general, it is fraught to infer function from microglial morphology alone, especially when doing so across different developmental stages.
- Methods, page 6, lines 246-252: Why were 3-way ANOVA’s not used to assess the interaction of CTL/ES, postnatal diet, and adult diet? This would be the most valid statistical approach, and SPSS can perform this type of test. Also, the post hoc test used, LSD, is equivalent to using multiple t-tests and does not include any correction for multiple comparisons, and thus is not a good choice. Please use a post hoc test that includes correction for multiple comparisons, such as Tukey’s post hoc test. The post hoc comparison of the CTL S-IMF STD and CTL S-IMF WSD groups would be valid if an appropriate post hoc test was used (not a t-test), following a significant interaction in the 3-way ANOVA.
- Discussion, page 15, lines 504-515: Your finding of increased Iba1 staining coverage in ES animals is associated with an increase in type 3 microglia (ramified), rather than amoeboid/activated microglia, at least in the S-IMF group. Thus, your claim that ES causes microglia to overreact to a later challenge (WSD) does not seem valid. If anything, this could be characterized as an “under-reaction” or resilience to the later challenge, rather than priming/sensitization. However, the issues with your interpretation again highlight the fact that the assessment of microglial numbers and morphology alone does not fully give insight into inflammatory response. Another measure, such as proinflammatory cytokine expression/production would be required to claim this, so please correct and temper your conclusions and clearly state the limitations of your measure of microglial number and morphology (which does not necessarily dictate function) in the discussion.
Round 2
Reviewer 1 Report
The authors have addressed all the comments/suggestions.
Author Response
We thank the reviewer for taking the time to read the manuscript and revisions we made, and are happy we addressed all comments/suggestions satisfactory.
On behalf of all authors, kind regards,
Silvie Ruigrok
Aniko Korosi